# An Online Survey Testing Factorial Invariance of the Optimization in Primary and Secondary Control Scales among Older Couples in Japan and the US

**DOI:** 10.3390/bs12110429

**Published:** 2022-11-01

**Authors:** Hideki Okabayashi, Akiko Takemura, Beth Morling

**Affiliations:** 1Department of Psychology, Meisei University, Tokyo 191-8506, Japan; 2Department of Psychology, Jin-Ai University, Fukui 915-8586, Japan; 3Department of Psychological and Brain Sciences, University of Delaware, Newark, DE 19716, USA

**Keywords:** OPS scale, factorial invariance, control strategies, cross-cultural comparison, subjective well-being

## Abstract

This study examines the factorial invariance of the Optimization in Primary and Secondary Control (OPS) scale and its associations with subjective well-being among older couples in Japan and the US. To this end, 200 older couples in Japan and 220 in the US were recruited through paid vendors and completed the questionnaire online. Couples were eligible if husbands were 70 years or older and wives were 60 years or older. A six-factor model, in which Compensatory Primary Control was subdivided into two factors, fit the data best; its factorial invariance was confirmed among the four subsamples. Compensatory Secondary Control was more strongly associated with subjective well-being in American couples than in Japanese couples, although the associations between well-being and the other five OPS factors were similar in the two countries. Future research on this six-factor model will be able to examine how these control strategies function in different cultures.

## 1. Introduction

Since Rothbaum et al. proposed a two-process model of perceived control over 40 years ago, researchers have studied how primary and secondary control correlate with well-being across both age and culture [1]. In primary control, people attempt to influence the immediate environment, outside themselves. Secondary control is directed inward, as people attempt to accommodate themselves to external realities. One important theory of primary and secondary control proposes that people optimize primary and secondary control processes depending on their age, situation, and cultural context. Specifically, as people age, secondary control is theorized to become dominant over primary control [2,3]. In addition, researchers have theorized that culture shapes people’s control preferences, with independent cultures emphasizing primary control and interdependent cultures emphasizing secondary control [4,5].

### 1.1. Measuring Primary and Secondary Control

For testing the aging hypothesis, one of the most widely used, theoretically derived measures of primary and secondary control is the Optimization in Primary and Secondary Control (OPS) scale [6]. The OPS scale consists of five factors: Optimization, Selective Primary Control, Compensatory Primary Control, Selective Secondary Control, and Compensatory Secondary Control [6].

In one study [7], Hasse et al. tested three self-report measures—the control scales of the OPS, Tenaciousness, and Flexibility (TenFlex) [8], and Selective Optimization with Compensation (SOC) [9]—together. They confirmed that three meta-factors exist: meta-regulation, goal engagement, and goal disengagement. The researchers also established that all three factors increase with age and are all associated with well-being.

However, few studies of the factor structure of the OPS scale itself have been conducted, and fewer have been performed using cross-cultural samples. The original study on scale development [6] did not provide information on the factor loadings of each factor’s corresponding items because parceling scores were used. In that study, each of the five factors in the OPS scale was constructed by three parceling scores, in which several item scores were aggregated [6]. Specifically, Heckhausen et al. divided 12 items of the Optimization factor into three parcels, each of which consisted of four items, and created three parceling scores by computing simple means of each set of four items [6]. Parceling scores were used in that case, because the maximum likelihood (ML) method cannot estimate the appropriate parameter values with ordinal data (the OPS response scales are ordinal). ML can, however, estimate appropriate parameter values when parceling scores are used, because parceling scores are considered continuous [10]. Given the increasing availability of methods for ordinal response scales, we can now analyze ordinal indicators directly with weighted least square estimation with robust standard errors and a mean- and variance-adjusted test statistic (WLSMV). In doing so, we can estimate the factor loadings of each item on its corresponding factors, something which the parceling method previously obscured.

### 1.2. Research Questions

The present study had four main research questions. First, we asked if we could establish factorial invariance of the five-factor OPS model across two cultures (Japan and the US) in a sample of older adults who were heterosexual married couples. If the original five-factor model did not fit the data, we planned to propose a more appropriate model of the OPS scale.

Second, we tested whether we could confirm factorial invariance among the four subsamples (Japanese men, Japanese women, US men, and US women).

Third, we examined gender and cultural differences among the factor scores.

Fourth, we examined associations of the factor scores with subjective well-being, including examining whether gender and culture moderated these associations. In this study, subjective well-being was operationalized according to Diener’s three-part definition, which measures satisfaction with life, positive affect, and negative affect [11].

## 2. Method

### 2.1. Samples

We contracted with vendors in Japan (*N* = 200 couples) and the US (*N* = 220) to recruit older adult married couples. In order to participate, all husbands needed to be at least 70 and wives at least 60 years old. Japanese couples had been married for an average of 50 years (because of an oversight, no data are available on the length of marriage in American couples). Online surveys were conducted in March 2018 in both countries. Mean ages were 78.15 (*SD* = 4.84) in Japanese husbands, 74.52 (*SD* = 5.55) in Japanese wives, 75.74 (*SD* = 4.54) in American husbands, and 71.43 (*SD* = 5.15) in American wives. In the US, ethnicity proportions of husbands and wives were as follows: 94.1% and 92.7% White, 1.4% and 1.8% Asian American, 1.8% and 2.3% Black, and 2.7% and 3.2% other or Latino.

Power analyses using G*Power Version 3.1.9.6. showed that given the current sample sizes (*N* = 220 in the US, *N* = 200 in Japan), *p* = 0.05, and an effect size of *r* = 0.20, the study’s statistical power was adequate, at 0.85 in the US and 0.81 in Japan.

### 2.2. Measurement

**OPS scale.** We administered the Optimization in Primary and Secondary Control scale (OPS) [6], which consists of 5 factors. We used the short version, whose 28-item were drawn from the original 44-item questionnaire based on their factor loadings in a previous survey by the first author [12]: Optimization (6 items), Selective Primary Control (6 items), Compensatory Primary Control (6 items), Selective Secondary Control (6 items), and Compensatory Secondary Control (4 items). The response scale ranged from 1 (*never true*) to 5 (*almost always true*).

**Subjective well-being.** According to Diener et al., subjective well-being (SWB) was measured with three components: satisfaction with life, frequency of positive affect, and frequency of negative affect [11]. The Satisfaction with Life Scale (SWLS) was measured with five items [13,14]. The SWLS has been used in multiple world cultures with meaningful results, suggesting that it is appropriate for use in cross-cultural research [15,16]. The scale ranged from 1 (*strongly disagree*) to 7 (*strongly agree*). Higher scores indicate higher satisfaction with life; the Cronbach’s *alpha* coefficient was 0.88, 93, 89, and 0.88, in Japanese husbands, Japanese wives, American husbands, and American wives, respectively.

Positive and negative affect were measured with eight items from the Positive and Negative Affect scales [17,18]. Participants were asked how often they felt each emotion during the last 30 days about four positive emotions (*cheerful, happy, peaceful, full of life*) and four negative emotions (*effortful, hopeless, restless or fidgety*, and *sad*). The scale ranged from 1 (*none of the time*) to 5 (*all of the time*). The Cronbach’s *alpha* coefficients were 0.81, 0.83, 0.90, and 0.89 for positive affect and 0.80, 0.82, 0.80, and 0.81 for negative affect in Japanese husbands, Japanese wives, American husbands, and American wives, respectively.

### 2.3. Analytic Procedure

We analyzed the data using packages and functions in the statistical software R, including the “psych” package for descriptive statistics and Cronbach’s alpha coefficients, the “anovakun” function for analyses of variance, and the “lavaan” package for confirmatory factor analysis (CFA). When conducting CFA, we used the WLSMV estimator for analyzing ordinal indicators and producing several goodness-of-fit indices such as χ^2^, Comparative Fit Index (CFI), Tucker–Lewis Index (TLI), and Root Mean Square of Error Approximation (RMSEA). The conventional levels for acceptable fit were as follows: CFI and TLI > 0.95 and RMSEA < 0.07 [19].

## 3. Results

### 3.1. Factor Structure of the OPS Scale

First, we performed ordinal CFA testing of the original five-factor model for all the participants in Japan and the US. There were 16 items in which no participants responded to the lowest category (*never true*) in at least one among the four subsamples. In these cases, we merged the *never true* category with the *seldom true* category because ordinal CFA cannot be executed when one of the response categories is empty. One item from the Compensatory Secondary Control scale was deleted because of its low factor loading. As a result, 27 items were analyzed. The robust goodness-of-fit indices did not meet conventional levels of acceptable fit (χ^2^ (314) = 6558.77, *p* < 0.001; CFI = 0.885, TLI = 0.871, RMSEA = 0.154; Table 1).

The modification indices suggested that, in the Compensatory Primary Control scale, there should be error correlations among two subsets of items: the three items CP3, CP5, and CP6, and the three items CP1, CP2, and CP4. The first three items seem to capture support seeking (“CP3. When I cannot solve a problem by myself, I ask others for help.”, “CP5. When difficulties become too great, I ask others for advice.”, and “CP6. When obstacles get in my way, I try to get help from others.”), while the remaining items seem to capture alternative strategies to compensate for lost primary control (“CP1. When I cannot get to a goal directly, I sometimes choose a roundabout way to achieve it.”, “CP2. When I can no longer make progress on something, I look for new ways to reach my goal.”, and “CP4. When obstacles get in my way, I find another way to get what I want.”). Therefore, we decided to divide the factor of Compensatory Primary Control into two subfactors: Support Seeking and Alternative Strategy.

The robust goodness-of-fit indices of this modified six-factor model were estimated (χ^2^ (309) = 2224.68, *p* < 0.001; CFI = 0.964, TLI = 0.960, RMSEA = 0.086) and a robust chi-square difference test showed that the fit of this six-factor model was significantly improved over that of the original five-factor model (Δχ^2^ (5) = 861.70, *p* < 0.001) (Table 1). The model improved significantly after we added one error correlation between two items in the factor of Optimization (“O1. It is important for me to be active not just in one area of life, but in several different ones.” and “O4. I stay active and involved in several different areas of life.”), which the modification index suggested were correlated, and whose meanings seem to be similar (Δχ^2^ (1) = 160.09, *p* < 0.001) and the robust goodness-of-fit met appropriate levels (χ^2^ (308) = 2033.16, *p* < 0.001; CFI = 0.968, TLI = 0.964, RMSEA = 0.082).

Each of the items of the OPS loaded highly (more than 0.66) on the factor specified by the theory (Table 2). However, there were two high correlations over 0.95: the correlation between Selective Primary Control and Selective Secondary Control was 0.958 and the correlation between Optimization and Selective Primary Control was 0.952 (Table 3). Therefore, it was necessary to examine the discrimination of these two factors. We used Bagozzi et al.’s method for testing factor discrimination, testing whether the correlation coefficient differs significantly from 1.00 or not, that is, whether (the correlation + 1.96*standard error) is greater than 1.00 [20]. The standard errors of the two correlation coefficients were 0.018 and 0.019 and the upper values of 95% confidence interval were 0.993 and 0.989, respectively, which were not greater than 1.00. The hypothesis that these two latent constructs were identical was rejected. Further, we examined whether merging these highly correlated factors made the fit better or not. In the descending order of size of factor correlations, the highest correlated two factors were merged into one and the model fit was compared successively (Table 4). When Selective Primary Control and Selective Secondary Control were merged, the robust goodness-of fit became significantly worse (Δχ^2^ (5) = 21.84, *p* < 0.001) and other goodness-of-fit indices also got worse in the five-factor model (CFI = 0.965, TLI = 0.961, RMSEA = 0.085). Therefore, all the successive merging processes made the fit significantly worse. We concluded that the six-factor model was the best one because these six latent constructs were statistically separate from each other and this model could explain the data most appropriately and parsimoniously (even though a few of the factor correlations were very high).

### 3.2. Factorial Invariance among Older Couples in Japan and the US

We compared the goodness-of-fit indices of the six-factor model with configural invariance (in which the factor structure is the same but no parameter was constrained) with those of the factorial invariance model (in which all the factor loadings equally constrained) among the four subsamples (Japanese husbands, Japanese wives, US husbands, US wives). Although a robust chi-square difference test showed that the factorial invariance model was significantly worse than the configural invariance model (Δχ^2^ (63) = 103.86, *p* < 0.001), other robust goodness-of-fit indices suggested that the factorial invariance model (CFI = 0.970, TLI = 0.968, RMSEA = 0.071) fit better than the configural invariance model (CFI = 0.964, TLI = 0.959, RMSEA = 0.080; Table 5). Although these fit indices were inconsistent, we decided to adopt the factorial invariance model. In addition to constraining factor loadings, when all the factor covariances were equally constrained, the correlation between Selective Primary Control and Selective Secondary Control was higher than 1.0 in Japanese husbands and this model was not identified. In the end, we adopted the factorial invariance model as a final one.

### 3.3. Gender and National Differences in These Factors’ Scores

Descriptive statistics and Cronbach’s *alphas* are shown in Table 6. We calculated these six-factor scores by taking the mean of the relevant items.

Gender and national differences in these factor scores were examined in mixed ANOVA. All six-factor scores were higher in American than in Japanese couples (all *p*s < 0.001) (Table 6). However, these mean differences between cultures should not be overinterpreted because they may simply be due to systematic differences in the way people in the two cultures use response scales [21]. In both countries, while Selective Primary Control, Alternative Strategy, and Selective Secondary Control were higher in husbands than in wives (*p* = 0.014, *p* = 0.009, and *p* = 0.027), Support Seeking was lower in husbands than in wives (*p* < 0.001).

### 3.4. Associations of the Six Control Factors with SWB

Pearson correlations of the 6 control factors with three aspects of SWB were calculated (Table 7). Three control factors, Optimization, Selective Primary Control, and Selective Secondary Control, were positively associated with all three aspects of well-being, while Support Seeking and Alternative Strategy were positively associated with the two aspects of well-being (SWLS and Positive Affect). When we examined whether these correlations were moderated by culture, we found significant differences between the two countries in the associations of Compensatory Secondary Control with SWB (Table 8), such that Compensatory Secondary Control was more strongly associated with SWB in American couples than in Japanese couples.

## 4. Discussion

This study examined the factorial invariance of the OPS scale in older couples in Japan and the US. We documented several interesting findings. First, using ordinal CFA on each item of the OPS scale instead of the parceling method, we proposed a six-factor model with one residual error correlation (the factor of Compensatory Primary Control) that was subdivided into Support Seeking and Alternative Strategy factors. This model fits the data better than the original five-factor model. In addition, each item has high factor loadings on its corresponding factor. Even though two pairs of factors were highly correlated with each other, the discrimination among these factors was statistically confirmed. This model also showed the best statistical fit compared with models with fewer factors.

Second, the factorial invariance of the six-factor model was confirmed among older couples in Japan and the US. Again, using ordinal CFA, we find that the overall framework in the OPS scale proposed by Heckhausen et al. [6] is generally maintained in the two different cultures, with the exception that the Compensatory Primary Control factor can be subdivided into two factors: Alternative Strategy and Support Seeking. This can lay the groundwork for further cross-cultural research.

Third, there were several gender differences in levels of control strategies in both countries. Selective Primary Control, Alternative Strategy (one of the two new factors), and Selective Secondary Control were higher in husbands than in wives, but Support Seeking (the other new factor) was higher in wives than in husbands. The finding seems consistent with traditional gender roles in which men are more likely to use agentic skills and abilities (Selective Primary Control and Alternative Strategy) and maintain motivation for a selected goal (Selective Secondary Control). In turn, women are encouraged to maintain social interactions, so they may be more likely to seek support from others. This pattern of results also complements other work which finds that in both the US and Japan, women are more likely to seek social support from others [22].

Fourth, we found that although the associations of control strategies with SWB were positive across the two cultures, there was one cultural difference. The association of Compensatory Secondary Control with SWB was significantly stronger in the US than in Japan. One aspect of this control strategy involves self-justification because, after failure, people remind themselves of their own effort or their own past accomplishments. Using this self-enhancing strategy may be more elaborated and approved in an individualistic cultural context (the US), more than in a collectivist cultural context (Japan). This pattern aligns with past research, in which college-aged Americans were more likely than Japanese to use self-esteem-enhancing strategies [23]. Our results suggest, then, that such cultural differences extend to older adults. Future research can replicate this finding in a new sample of older adults, and a broader range of self-enhancement measures.

One strength of this study was that it tested older adults in two cultures. While much research on the OPS has compared older, middle-aged, and younger adults, very little has tested cultural differences in the OPS.

There were several limitations in this study. First, the samples provided by vendors in both countries were not random samples of their respective populations. They were biased to include older adults who are willing to seek out paid surveys.

Second, the US sample is almost entirely White, so any conclusions are limited to this subgroup. In American culture, White American contexts are probably the most likely to foreground individualism and independence. Therefore, if anything, our primarily White American sample was biased to find more, rather than less, cultural differences when compared with Japan. In this context, it is notable that we actually found few differences between this White American sample and a Japanese sample.

A third limitation is that although we confirmed the factorial invariance of a six-factor model in older couples in Japan and the US, the factor structure should be reconfirmed before assuming it would apply to additional cultures. In particular, careful attention should be paid to the highly correlated factors.

We did find several positive associations, all of which are consistent with the argument that both primary and secondary control strategies are associated with well-being in both the US and Japan. However, a limitation is that our cross-sectional design did not allow us to determine the causal direction between control strategies and SWB.

In conclusion, we provide a modified six-factor model of the OPS scale, which fits the data better than the original five-factor model. This six-factor model will enable future researchers to examine cross-cultural differences in, and well-being correlates of, these well-known primary and secondary control scales.

## Figures and Tables

**Table 1 behavsci-12-00429-t001:** Comparison of the robust goodness-of-fit indices of configural invariance models among four samples.

*N* of Factors	Model Variant	χ^2^	*df*	*p*	CFI	TLI	RMSEA	Δχ^2^	Δ*df*	Δ*p*
5	The original five-factor model	6558.77	314	<0.001	0.885	0.871	0.154			
6	Compensatory Primary control was divided into two factors (Alternative Strategy and Support Seeking).	2242.68	309	<0.001	0.964	0.960	0.086	−861.70	−5	<0.001
6	Compensatory Primary control was divided into two factors (Alternative Strategy and Support Seeking) and added one error correlation.	2033.16	308	<0.001	0.968	0.964	0.082	−160.09	−1	<0.001

**Table 2 behavsci-12-00429-t002:** Standardized factor loadings of Optimization in Primary and Secondary Control Scale.

Optimization		Factor Loadings
O1	It is important for me to be active not just one area of life, but in several different ones.	0.73
O2	It is important for me that a new goal can be pursued over the long term.	0.79
O3	I pursue new goals when the time is right for me.	0.85
O4	I stay active and involved in several different areas of life.	0.78
O5	I invest my time in developing broad skills that can be used in many areas.	0.79
O6	I choose goals that have more long-term as opposed to short-term benefits.	0.68
Selective Primary Control		
SP1	When I have a goal, I am willing to work hard at sharpening the skills in order to achieve it.	0.83
SP2	When I really want something, I am able to work hard to achieve it.	0.86
SP3	When obstacles get in my way, I put in more effort.	0.89
SP4	When I have set a task for myself, I try to learn the skills necessary to do it well.	0.89
SP5	Once I have decided on a goal, I do whatever I can to achieve it.	0.85
SP6	When a goal is more difficult than expected, I try harder to achieve it.	0.90
Alternative Strategy		
CP1	When I cannot get to a goal directly, I sometimes choose a roundabout way to achieve it.	0.72
CP2	When I can no longer make progress on something, I look for new ways to reach my goal.	0.86
CP4	When obstacles get in my way, I find another way to get what I want.	0.89
Support Seeking		
CP3	When I cannot solve a problem by myself, I ask others for help.	0.86
CP5	When difficulties become too great, I ask others for advice.	0.95
CP6	When obstacles get in my way, I try to get help from others.	0.96
Selective Secondary Control		
SS1	When I have chosen a difficult task for myself, I imagine how proud I will be when I have solved it.	0.79
SS2	When I have decided on something, I know that I will achieve it.	0.80
SS3	When I have decided on something, I always remind myself that it was the right decision.	0.68
SS4	Once I decide on something, I am not easily distracted by other things.	0.73
SS5	When I have set a goal for myself, I keep in mind that I also have the abilities to achieve it.	0.86
SS6	When I have decided on a goal, I always keep in mind its benefits.	0.83
Compensatory Secondary Control		
CS1	When I get into a difficult situation, I remind myself that in many ways I am better off than other people.	0.81
CS2	When I have not accomplished something important, I console myself by thinking about other areas where I had more success.	0.66
CS3	When I doubt myself, I keep in mind that I have already accomplished a lot in my life.	0.76

Notes. A residual correlation between O1 and O4 was 0.461 (*p* < 0.001). The robust goodness-of-fit indices are χ^2^ (308) = 2033.161, CFI = 0.968, TLI = 0.964, RMSEA = 0.082.

**Table 3 behavsci-12-00429-t003:** Factor correlations.

	1	2	3	4	5
1. Optimization					
2. Selective Primary Control	0.952				
3. Alternative Strategy	0.893	0.901			
4. Support Seeking	0.457	0.378	0.436		
5. Selective Secondary Control	0.924	0.958	0.859	0.429	
6. Compensatory Secondary Control	0.807	0.678	0.766	0.511	0.795

**Table 4 behavsci-12-00429-t004:** Comparison of robust goodness-of-fit indices of configural invariance model in modified 6-factor model and combined 5-factor model.

Models	χ^2^	*df*	*p*	CFI	TLI	RMSEA	Δχ^2^	Δ*df*	Δ*p*
6 factor model with one error correlation	2033.16	308	<0.001	0.968	0.964	0.082			
5 factor model: SPC + SSC	2188.43	313	<0.001	0.965	0.961	0.085	21.84	5	<0.001
4 factor model: SPC + SSC + OPT	2316.70	317	<0.001	0.963	0.959	0.087	114.08	4	<0.001
3 factor model: SPC + SSC + OPT + ALT	2432.60	320	<0.001	0.961	0.957	0.089	93.13	3	<0.001
2 factor model: SPC + SSC + OPT + ALT + CSC	2827.87	322	<0.001	0.954	0.950	0.096	40.85	2	<0.001
1 factor model: SPC + SSC + OPT + ALT + CSC + SUP	9083.31	323	<0.001	0.838	0.824	0.180	567.66	1	<0.001

Note. SPC = Selective Primary Control; SSC = Selective Compensatory Control; OPT = Optimization; ALT = Alternative Strategy; CSC = Compensatory Selective Control; SUP = Support Seeking.

**Table 5 behavsci-12-00429-t005:** Robust goodness-of-fit indices with modified 6-factor models in 2 levels of invariance among 4 subsamples.

Models	χ^2^	*df*	*p*	CFI	TLI	RMSEA	Δχ^2^	Δ*df*	Δ*p*
Configural invariance model	2896.27	1232	<0.001	0.964	0.959	0.080			
Factorial invariance model	2658.23	1295	<0.001	0.970	0.968	0.071	103.86	63	<0.001

**Table 6 behavsci-12-00429-t006:** Descriptive statistics and *F* values of mixed ANOVA of Nation by Gender on 6 OPS scale scores.

		JAPAN (*N* = 200)			US (*N* = 220)			*F* (1, 418)				
		*M*	*SD*	α	*M*	*SD*	α	Nation		Gender		Nation × Gender
Optimization	Husbands	3.16	0.69	0.86	3.65	0.73	0.87	85.68 ***	US > JP	0.82		0.71
	Wives	3.10	0.68	0.87	3.65	0.65	0.85					
Selective Primary Control	Husbands	3.47	0.77	0.90	3.98	0.74	0.93	86.48 ***	US > JP	6.11 *	H > W	1.90
	Wives	3.32	0.74	0.92	3.94	0.66	0.91					
Alternative Strategy	Husbands	3.21	0.69	0.79	3.67	0.76	0.81	77.44 ***	US > JP	6.81 **	H > W	1.75
	Wives	3.06	0.66	0.80	3.62	0.72	0.80					
Support Seeking	Husbands	2.64	0.72	0.85	3.42	0.87	0.91	143.72 ***	US > JP	31.55 ***	H < W	0.02
	Wives	2.90	0.81	0.90	3.67	0.80	0.86					
Selective Secondary Control	Husbands	3.33	0.67	0.85	3.78	0.67	0.88	72.16 ***	US > JP	4.94 *	H > W	0.74
	Wives	3.25	0.65	0.87	3.72	0.63	0.85					
Compensatory Secondary Control	Husbands	2.82	0.68	0.66	3.52	0.75	0.71	116.36 ***	US > JP	2.47		3.02
	Wives	2.94	0.68	0.70	3.51	0.72	0.73					

Note. US = American Sample; JP = Japanese Sample; H = Husbands; W = Wives. * *p* < 0.05, ** *p* < 0.01, *** *p* < 0.001.

**Table 7 behavsci-12-00429-t007:** Pearson’s Correlations between OPS factor scores and Well-beings.

	SWLS	Positive Affect	Negative Affect
Optimization	0.51 ***	0.51 ***	−0.14 ***
Selective Primary Control	0.47 ***	0.46 ***	−0.15 ***
Alternative Strategy	0.38 ***	0.33 ***	−0.05
Support Seeking	0.34 ***	0.22 ***	0.03
Selective Secondary Control	0.49 ***	0.47 ***	−0.16 ***
Compensatory Secondary Control	0.41 ***	0.34 ***	−0.02

Note. SWLS = Satisfaction With Life Scale. *** *p* < 0.001.

**Table 8 behavsci-12-00429-t008:** National differences in correlations between Compensatory Secondary Control and Well-being.

	JAPAN	US	Δχ^2^ (1)
SWLS	0.17 ***	0.39 ***	14.39 ***
Positive Affect	0.22 ***	0.39 ***	5.78 *
Negative Affect	0.09	−0.14 **	11.44 ***

Note. SWLS = Satisfaction With Life Scale. * *p* < 0.05, ** *p* < 0.01, *** *p* < 0.001.

## Data Availability

Not applicable.

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
