# Peer review of "An Online Survey Testing Factorial Invariance of the Optimization in Primary and Secondary Control Scales among Older Couples in Japan and the US"

_behavsci, 2022, doi:10.3390/bs12110429_

Round 1

Reviewer 1 Report

Thank you for giving me to review your manuscript. This manuscript is interesting and scientifically meaningful for considering the optimization in primary and secondary control scales among older couples in Japan and the US. Regarding the contents, the following revision should be considered.

The title should include the study design.

Reference and affiliation formats have flaws and should be revised comprehensively.

There are some long paragraphs. The author should focus on theory building, the problems, and research question paragraphs. The first to third paragraphs contain mixed contents. The first paragraph should focus on the general information regarding the optimization of primary and secondary control. Moreover, the second and third paragraphs should introduce the research questions as the theoretical and conceptual framework, including research questions.

The introduction should include the research question of this study.

The sample section of the method contains no descriptions regarding sample calculation.

The discussion part should be based on paragraph writing. There are too long paragraphs, and they are not friendly for readers.

This study should describe the limitation of sampling bias and the results' applicability to other settings, and the future investigation in the limitation part.

In the conclusion or discussion, the study’s strengths should be focused on international readers.

The conclusion should be modest. This study has many limitations and should describe just a possibility.

Author Response

Responses to Reviewer 1

Thank you very much for your comments to our manuscript. We addressed each of your comments as follows.

1. The title should include the study design.

We changed the title to include study design:

An Online Survey Testing Factorial Invariance of the Optimization in Primary and Secondary Control Scales Among Older Couples in Japan and the US

2. Reference and affiliation formats have flaws and should be revised comprehensively.

We arranged all the reference and affiliation formats according to this journal.

3. There are some long paragraphs. The author should focus on theory building, the problems, and research question paragraphs. The first to third paragraphs contain mixed contents. The first paragraph should focus on the general information regarding the optimization of primary and secondary control. Moreover, the second and third paragraphs should introduce the research questions as the theoretical and conceptual framework, including research questions. The introduction should include the research question of this study.

We reorganized the Introduction into concise paragraphs. For example, we divided the introduction into three subsections: general information, "Measuring Primary and Secondary Control," and "Research Questions."

4. The sample section of the method contains no descriptions regarding sample calculation.

We added information about our sample size calculations as follows:

"Power analyses using G*Power Version 3.1.9.6. showed that given the current sample sizes (N = 220 in the US, N = 200 in Japan), p= .05, and an effect size of r = .20, the study’s statistical power was adequate, at .85 in the US and .81 in Japan."

5. The discussion part should be based on paragraph writing. There are too long paragraphs, and they are not friendly for readers.

As with the introduction, we broke up the paragraphs in the discussion so that each limitation was discussed in its own paragraph

6. This study should describe the limitation of sampling bias and the results' applicability to other settings, and the future investigation in the limitation part.

In addition to a statement we already had in our manuscript ("They may be biased to include older adults who are more willing to seek out paid surveys"), we also added the following limitation regarding our sample composition: "the US sample is almost entirely White, so any conclusions are limited to this subgroup. In American culture, White American contexts are probably the most likely to foreground individualism and independence. If anything, our primarily White American sample was biased to find more, rather than less, cultural difference when compared to Japan. Therefore, it is notable that we actually found few differences between this White American sample and a Japanese sample."

7. In the conclusion or discussion, the study’s strengths should be focused on international readers.

We added one paragraph for the study's strengths in the discussion as follows:

“One strength of this study was that it tested older adults in two cultures. While much research on the OPS has compared older, middle-age, and younger adults, very little has tested cultural differences.”

8. The conclusion should be modest. This study has many limitations and should describe just a possibility.

We adjusted the wording in several places in the discussion to balance our statements about the study's contributions with appropriate qualifications and limitations.

Reviewer 2 Report

This is an important research concerning a specific tool of measurement. However, your article must improve and go to another level. First the language, spelling, etc., must be improved, i saw three spelling, grammar errors only in the abstract. So, please have a native english reviewer to go through the words in detail. The sample is really an all white sample, which is almost unacceptable today, unless it is part of the study, you must explain why your sample is so heavily dominated, and your paid researchers did not work hard enouogh to gain a sample of the population. you must account for this. And lastly, what does this really tell us about aging, about improving the lives of older adults. Your conclusion should be must longer and compelling. 

Author Response

Responses to Reviewer 2

Thank you very much for your comments to our manuscript. We addressed each of your comments as follows.

1. First the language, spelling, etc., must be improved, I saw three spelling, grammar errors only in the abstract. So, please have a native English reviewer to go through the words in detail.

One of the authors, who is a native English speaker, checked and edited the entire document.

2. The sample is really an all white sample, which is almost unacceptable today, unless it is part of the study, you must explain why your sample is so heavily dominated, and your paid researchers did not work hard enough to gain a sample of the population. you must account for this.

When we addressed the limitations of our sampling technique, we added the following paragraph:

"Second, the US sample is almost entirely White, so any conclusions are limited to this subgroup. In American culture, White American contexts are probably the most likely to foreground individualism and independence. If anything, our primarily White American sample was biased to find more, rather than less, cultural difference when compared to Japan. Therefore, it is notable that we actually found few differences between this White American sample and a Japanese sample."

3. And lastly, what does this really tell us about aging, about improving the lives of older adults. Your conclusion should be must longer and compelling.

Throughout the revised Discussion, we attempted to articulate our contributions and conclusions in a way that balances the study's contributions and limitations, without overstating the significance of our data.

Round 2

Reviewer 1 Report

The manuscript has been considerably improved. I think that this paper is suited for inclusion in our journal.